# Epigenome Chaos: Stochastic and Deterministic DNA Methylation Events Drive Cancer Evolution

**DOI:** 10.3390/cancers13081800

**Published:** 2021-04-09

**Authors:** Giusi Russo, Alfonso Tramontano, Ilaria Iodice, Lorenzo Chiariotti, Antonio Pezone

**Affiliations:** 1Dipartimento di Medicina Molecolare e Biotecnologie Mediche, Università di Napoli “Federico II”, 80131 Naples, Italy; giusi.russo2@studenti.unina.it (G.R.); ilaria.iodice2@studenti.unina.it (I.I.); chiariot@unina.it (L.C.); 2Department of Precision Medicine, University of Campania “L. Vanvitelli”, 80138 Naples, Italy; alfonso.tramontana@unicampania.it

**Keywords:** genomic instability, epigenetic alterations, clonal expansion, clonal selection, genome and epigenome chaos, cancer evolution

## Abstract

**Simple Summary:**

Cancer is a group of diseases characterized by abnormal cell growth with a high potential to invade other tissues. Genetic abnormalities and epigenetic alterations found in tumors can be due to high levels of DNA damage and repair. These can be transmitted to daughter cells, which assuming other alterations as well, will generate heterogeneous and complex populations. Deciphering this complexity represents a central point for understanding the molecular mechanisms of cancer and its therapy. Here, we summarize the genomic and epigenomic events that occur in cancer and discuss novel approaches to analyze the epigenetic complexity of cancer cell populations.

**Abstract:**

Cancer evolution is associated with genomic instability and epigenetic alterations, which contribute to the inter and intra tumor heterogeneity, making genetic markers not accurate to monitor tumor evolution. Epigenetic changes, aberrant DNA methylation and modifications of chromatin proteins, determine the “epigenome chaos”, which means that the changes of epigenetic traits are randomly generated, but strongly selected by deterministic events. Disordered changes of DNA methylation profiles are the hallmarks of all cancer types, but it is not clear if aberrant methylation is the cause or the consequence of cancer evolution. Critical points to address are the profound epigenetic intra- and inter-tumor heterogeneity and the nature of the heterogeneity of the methylation patterns in each single cell in the tumor population. To analyze the methylation heterogeneity of tumors, new technological and informatic tools have been developed. This review discusses the state of the art of DNA methylation analysis and new approaches to reduce or solve the complexity of methylated alleles in DNA or cell populations.

## 1. Introduction

Uncontrolled cell growth is the dominant characteristic of cancer cells and is due to genomic instability and epigenetic alterations [1]. The genomic instability as increased tendency to accumulate mutations includes small (single nucleotide variants, insertion, deletion, and microsatellite instability) and large structural variations (aneuploidy and chromosome instability) [2]. The epigenetic changes are characterized by addition or removal of chemical groups (mainly methyl groups) to the DNA or histones, which wrap DNA in chromosomes and can alter gene expression and chromatin structure [3]. The heterogeneity between tumors (inter-tumor) and within tumors (intra-tumor) depends on these genetic and epigenetic alterations [4]. Since the genetic changes are very rapid and frequent over time, the use of genetic markers in unstable cancer cells is not informative as far as the tumor evolution is concerned. On the other hand, also DNA methylation shows a high inter-tumor and intra-tumoral heterogeneity and is defined globally as aberrant methylation [5,6], due to the high polymorphism of the somatic methylation traits in all chromosomes [7,8,9]. We refer to these epigenetic changes as “epigenomic chaos” because they are generated by a mix of random and deterministic events, which are not easily separable. The evolution of cancer clones can be explained by the chaos theory, a concept borrowed from non-linear dynamic systems, which evolve with apparent randomness but contain “underlying patterns, interconnectedness, constant feedback loops, repetition, self-similarity, fractals, and self-organization” [10].

In fact, methylation changes during tumor evolution are aperiodic, repetitive, and very sensitive to the initial states of the system (e.g., cell type or differentiation state). The genetic and epigenetic changes in evolving cancer cells are randomly generated but are selected by deterministic events (e.g., chemoresistance). However, at present, we do not know whether aberrant methylation is the cause or the consequence of cancer progression.

To analyze the epigenetic heterogeneity of tumors, new technological and informatic tools have been developed. This review discusses the state of the art of the epigenomic analysis and new approaches to dissect the complexity of cancer epigenomes.

## 2. Cancer Evolution

Cancer evolution is characterized by the accumulation of mutations and epimutations (e.g., variations in CpGs methylation status) in somatic cells. At beginning of this evolution neoplasms are clusters of cells that display the same genetic and epigenetic profiles. With the time, new genetic and epigenetic variants emerge and are transmitted to daughter cells, which can acquire new genetic and epigenetic anomalies. Competition for space and resources, increased survival, and reproduction success may depend on mutations and epimutations leading to increased fitness of clones in the population of cells [11] (Figure 1). Anti-cancer therapies can also induce selection of clones, killing sensitive cells, but leaving resistant cells behind [12]. Cancer is an example of Darwinian evolution and selection at the genetic and epigenetic levels. For example, methylation and downregulation of CDKN2A gene (CDK4 and 6 inhibitor) permit the exit from senescence induced by oncogenes [13]. Cells bearing both methylated CDKN2A and protooncogene mutations acquire a greater fitness compared to cells with the oncogene mutations without CDKN2A-B epigenomic changes. Cells with both changes have a selective advantage that will be more evident in a hostile environment (hypoxia, therapies, and multilevel metastasis process) [14,15,16,17].

Cancer evolution can be separated into two evolutionary phases: macro-evolutionary in which genomic instability generates new genetic and epigenetic assets (clones) and a micro-evolutionary phase in which the cells transmit the genetic and epigenetic modifications to daughter cells stabilizing these variants. Fitness selection of genetic and epigenetic changes occurs mainly within the micro-evolutionary phase (deterministic evolution) [18].

## 3. Genome Instability (Genome Chaos)

We believe that evolution of cancer cell clones is shaped initially by randomness and eventually by fitness selection. Genomic instability is a critical hallmark of cancer. Such instability derives from genetic or epigenetic alterations acquired during cancer progression and helps selected cancer cells to be competitive for survival [18,19]. These modifications may underlie alterations of the cell cycle and resistance to programmed cell death [19,20]. Both endogenous genotoxic stress (such as reactive oxygen species (ROS) resulting from cell metabolism, DNA replication, or transcription) and exogenous genotoxic insults (such as ultraviolet light, ionizing radiation, or chemicals harmful to DNA) can induce genetic alterations leading to genomic instability.

The study of genomic instability has revealed the complexity of clonal evolution characterized by chaotic and continuously evolving genomes (accumulation of genetic alterations) [21]. Not all genetic alterations will endure. In fact, many of them will be eliminated during a phase of adaptation and fixation of the mutation until a partial stability is reached [22,23,24]. This strategy allows tumor cells to employ a stepwise adaptative response to environmental cues [21,25,26]. These unstable and chaotic states increase the complexity of tumor cell populations when considered over time [27,28,29,30,31]. Somatic mutations can also accumulate with aging and contribute to cancer [32,33] supporting the conclusion that there is a linear relationship between age and number of mutations [34,35,36,37]. Cancer-associated mutations have been found in a variety of normal tissues of individuals of all ages [38,39,40], including bone marrow [41,42,43], esophagus [44,45], colon [46], brain [47], endometrium and gynecological tissues [48,49,50,51,52], and a comprehensive set of 29 human tissues [35,53]. The presence of mutations in leukemia driver genes is a phenomenon known as clonal hematopoiesis [54] and concerns the 3% of healthy individuals under 30 [55] and 95% of individuals aged over 50 [48]. These mutations affect genes or loci implicated in cancer and are associated with diffuse inflammation [56,57,58,59,60]. The most frequently mutated genes are DNMT3a, TET2, and ASXL, which are powerful epigenetic regulators. Although the mutations in these genes are not accompanied by hematological disorders, they can trigger disordered epigenetic modifications, which can amplify silent genomic changes [61]. These findings are rapidly expanding and reshaping our understanding of human carcinogenesis and aging. In addition, the immune system applies a further selective pressure on cancer clone evolution. In fact, one of the important functions of the immune system is to recognize and eliminate cancer or senescent cells. It is plausible that aging of the immune system (senescence) can reduce the selective pressure and favor the amplification of mutated clones [59,61]. One general conclusion derived from these data is that the presence or the accumulation of mutations in growth or survival-controlling genes (oncogenes and suppressors) does not induce per se genome instability in cells that maintain the epigenetic memory. Furthermore, these findings reduce the power of the genetic markers to accurately predict tumor evolution.

## 4. Epigenome Chaos

Cancer cells show also epigenetic alterations, referred as aberrant methylation [62,63]. The organization of chromatin, as the topologically associating domains (TAD) and gene expression profiles in normal cellular become unrecognizable in cancer cells. A dense hypermethylation of the CpG islands and a hypomethylation of regulatory regions (such as centromeres) characterize the genome of transformed cells [63]. These dramatic changes may lead to chromosomal instability and may contribute to transcriptional silencing of tumor suppressor genes [64]. Promoter hypermethylation of genes regulating apoptosis (DAPK and APAF-1), cell cycle (p16INK4a, p14ARF, and p15INK4b), and DNA repair (hMLH1, BRCA1, and MGMT) has been found in many cancers. The epigenetic aberrations observed in cancer can be summarized as follows: (1) transcriptional silencing of tumor suppressor genes by CpG island promoter hyper-methylation [64,65], (2) global genomic hypomethylation [66], (3) loss of imprinting (LOI) [65,67], (4) loss of epigenetic repression of intragenomic “parasites” [68], and altered expression of homeobox genes [68]. These changes contribute to the evolution of cancer clones by balancing the activation of oncogenes that would otherwise induce senescence [69].

However, CpG hypermethylation is not always associated to silencing. A new notion on the impact of methylation on gene expression is emerging. The inhibition or activation of transcription by methylation is dependent on the gene segment analyzed. For example, hypermethylation at gene bodies is associated with gene expression, while hypermethylation of TSS or enhancers leads invariably to silencing. This is relevant in pan cancer methylome analysis because the activation of homeobox genes in many cases is due to hypermethylation of the gene body (Figure 2).

Conversely, methylation at the enhancer sites leads to gene silencing [13]. We conclude that aberrant CpG island methylation can be used as a biomarker of evolution of cancer cells and represents a possible target for therapies [10,11,12,13,69,70,71,72,73,74].

## 5. Mechanisms of de Novo DNA Methylation

The epigenetic modifications change the structure of chromatin, its accessibility, and, ultimately, modify gene expression. The epigenomic chaos in cancer, which generates various patterns of hypermethylated and hypomethylated loci, has stimulated a debate on the evolution of neoplasms. In particular, a question still unanswered is whether the initial processes that induce epigenetic, specifically, methylation, changes are stochastic or deterministic, i.e., they are driven by random events that change the methylation status of any segment of DNA or by factors that target specific DNA regions and change the local methylation profile(s). There are many examples showing that targeting specific factors changes the methylation status of specific DNA segments. For example, gene products that silence the INK4-ARF tumor suppressor locus in a human colorectal cancer cell line have been reported [75]. The mutated KRAS protein in the same cell line inhibits the degradation and stabilizes the transcription factor, ZNF304, which forms a co-repressor complex containing a DNA methyltransferase that induces de novo DNA methylation at the INK4-ARF locus resulting in inhibition of transcription. Similarly, we have shown, in two different experimental models, that DNA damage induces the silencing of the soluble Wnt inhibitor WIF1 via ATM [76] and that TGF-β1 induces activation and repression of target genes by modifying the conformation of the chromatin at the promoter sites [77]. On the other hand, there are also examples of stochastic or random de novo methylation due to damage (Double Strand Breaks, DSB) and repair (Homologous Recombination, HR) (see below) or incomplete demethylation, C hydroxymethylation, or contiguity of methylated traits (Figure 3).

However, it is worth to note that the transmission of epigenetic features is a mechanism underlying embryonic development that we know with the term imprinting [78], which regulates the expression of some loci based on the origin of the allele (maternal or paternal). The transmission of a methylated locus from the mother cell to the daughter cell is ensured by the enzyme DNA methyl-transferase 1 (DNMT1). The activity of DNMT1 in maintenance of DNA methylation is supported by its substrate preference for hemi-methylated CpG sites, as well as by high enzymatic processivity [79]. We have demonstrated with a synthetic gene (GFP) that DNA damage and HR induce strand-specific (allele-specific) methylation that is transmitted to the offspring generating undermethylated high expressor clones and hypermethylated low expressor cells [8,9,10,11,65]. In addition, we have shown that transcription in a short time window (15 days after DSB and HR) edits further local methylation, which is stably transmitted to daughter cells and can be monitored overtime in evolving cell populations [7,8,9,70]. The transmission of the somatic epigenetic traits from mother to the daughter cells allows tumors to use this strategy to generate a great variety of clones and select epigenetically permissive clones, for example, clones in which the DNA damage-HR resulted in de novo methylation and silencing of a tumor suppressor. This trait confers an obvious advantage to the clone, because it can escape oncogene-induced senescence [8,9,10].

## 6. New Tools for DNA Methylation Analysis

Before discussing the new methods of DNA methylation analysis, it may be useful to remark some concepts and definitions reported here. A clone is defined as a group of tumor cells that shares the same mutational profile, while a subclone is a group of tumor cells that originates from the clone and diverges by acquiring new additional mutations [80,81]. A cluster is a group of cells that share a fraction of the mutational profiles. Another important concept is fitness, which refers to the ability of a tumor cell to survive and proliferate. Increased fitness can lead to clonal expansions characterized by expansion of one genotype in the tumor cell populations (Figure 1) [81,82].

A large degree of variation in the number of subclones at both the genetic and epigenetic levels is detected in human cancers, although the relative relevance of this phenomenon is influenced by technical reproducibility and sequencing depth. However, two key principles are relevant: 1. evolution time and 2. presence or absence of intermediates during evolution. However, we note that (1) the analysis of the mutations of cancer driver genes alone is complicated by the epigenetic disorder (chaos) and genetic instability that drive tumor heterogeneity inter- and intra-tumor heterogeneity and (2) the different methods of methylation analysis that identify differentially methylated cytosines (DMC) or differentially methylated regions (DMR) show extreme heterogeneity and polymorphism and are not informative on the tumor progression [5,6,7,8,9,76,83] (see Figure 3 and Figure 4). Below are reported the current methods used for DNA methylation analysis.

## 7. Epialleles-Based Analysis (EBA)

These methods are based on the concept that clones derived from a single positively selected progenitor will have a unique epigenetic configuration (epialleles). In particular, the Epialleles-Based Analysis (EBA) identifies methylated DNA molecules with defined 5’ and 3’ ends (epihaplotype) generating a binary profile (0 unmethylated/1 methylated) of CpGs in DNA strings (Figure 4). This method generates binary matrices (0 and 1)/CpG/locus. This binary matrix provides information about the constitution and complexity of the populations considering that 1 cell = 1 epiallele.

The tools based on EBA are: 1 *AmpliMethProfiler* [84] (used for single loci), 2. MethCoresProfiler [83], and 3. *Methclone* [85] (used to analyze genome-wide methylomes (RRBS and WGBS)).

AmpliMethProfiler, a python-based pipeline, extracts and performs statistical epihaplotype analysis of amplicons from targeted deep bisulfite sequences. This tool investigates the methylation diversity by directly extracting the methylation profiles (epihaplotypes) at a single locus in the sequence population [84,86]. Using this high-throughput approach, the epihaplotypes can be treated as haploid organisms with a specific frequency in the population (Shannon Entropy).MethCoresProfiler, a R-based pipeline, traces and tracks CpGs in the same phase (methylated or not methylated cores) shared by families of epialleles by calculating their frequency in the population (MethCore Index), the frequency normalized to the mean methylation (Clonality Index), and the association index between the CpGs belonging to the same core normalized to the average methylation of the population of sequences (Entanglement Index) [83]. This tool is able to recognize the original epigenetic ancestor from which the molecules of different epialleles derive, considering each addition or removal of a methyl groups as independent events. This method allows the reconstruction of the evolution of families of epialleles from a common ancestor. Note that the frequency of individual epialleles is usually not statistically significant, while the frequency of the common signature (core) is significant. This tool analyses amplicons from targeted deep bisulfite sequencing and allows the analysis of several samples longitudinally.Methclone extracts and performs statistical epihaplotype analysis for each locus from genome-wide DNA NGS data (RRBS and WGBS). It is based on the comparison between two samples longitudinally and identifies the epigenetic loci hosting large clonal variations. It quantifies epiallele shift(s), as the Hamming distance and the frequency of single epialleles [85].

## 8. Conclusions

Cancer is an ever-changing disease characterized by both genetic and epigenetic dynamic alterations. Although mutations in driver genes are necessary for the proliferation of a clone, numerous other mutations are found in tumor cells [29,30]. Furthermore, the mutations in the driver genes can also be found in healthy subjects (clonal hematopoiesis) linked to cellular aging or to immunological senescence [35,38,39,40,41,42,43,44,45,46,47,48,49,50,51,52,53,54]. The accumulation of gene mutations and chromosomal alterations, the high intra- and inter-tumor heterogeneity, the fluctuations in the frequency of the mutations in the various stages of neoplastic evolution generate a dynamic pattern of evolution of the cell populations very similar to the chaotic evolution defined in dynamic physical systems. In the Figure 1 is reported a model of cancer evolution in which clones acquire different mutations over time under environmental pressure, whereas DNA methylation is passed almost intact on to all offspring. This condition balances oncogene-induced senescence. Further evolution of cancer cell populations is induced by epigenetic chaos, which involves disordered hyper and hypomethylation in various DNA segments, leading to the redistribution of repressor sites and loss of epigenetic memory, which we call aberrant methylation. We believe that the epigenetic chaos unleashes genomic instability by promoting expression of fetal genes and silenced transposons [87]. These features characterize the late-stage therapy-resistant cancers (Figure 1). The complexity of methylation profiles in the unstable cancers cannot yet be analyzed with the current methods. New tools need to be developed and validated, and recently, new approaches based on analytic systems borrowed from metagenomics and population genetics [83,84,85] are being tested. We believe that this approach represents an important tool to decipher the methylation profiles in complex populations.

## Figures and Tables

**Figure 1 cancers-13-01800-f001:**
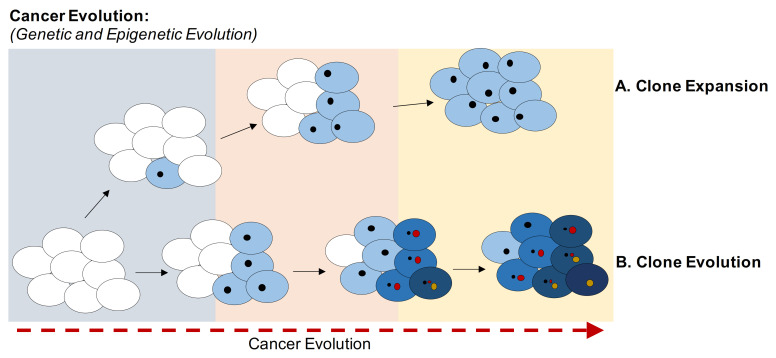
Evolution of cancer clones. Schematic representation of the evolution of clones based on the accumulation of genetic and epigenetic variants. Cancer cells acquire different mutations over time under environmental pressure, whereas DNA methylation is passed on to the offspring to balance oncogene-induced senescence. (**A**) Schematic representation of the expansion of a clone in which all cells share same genetic and epigenetic variants. (**B**) Schematic representation of the evolution of clones based on the acquisition of new variants. The higher fitness is shown as a darker tone. Each clone acquires different genetic and epigenetic variants, randomly generated but selected by the microenvironment and DNA-damaging therapies.

**Figure 2 cancers-13-01800-f002:**
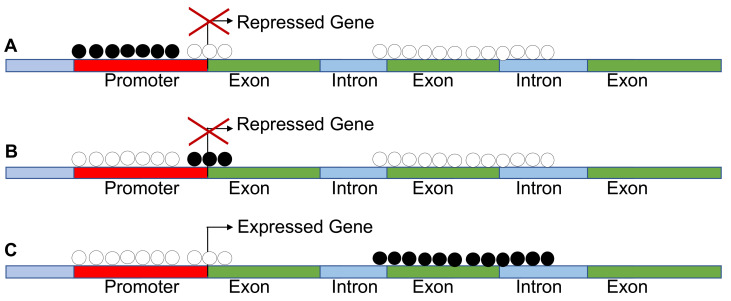
Methylation is not always associated with silencing of gene expression. (**A**–**C**) These show that methylation of the promoters and transcription start sites (TSS) leads to repression of gene expression, while methylation of the gene body results in increased expression as found in pan cancer methylomes.

**Figure 3 cancers-13-01800-f003:**
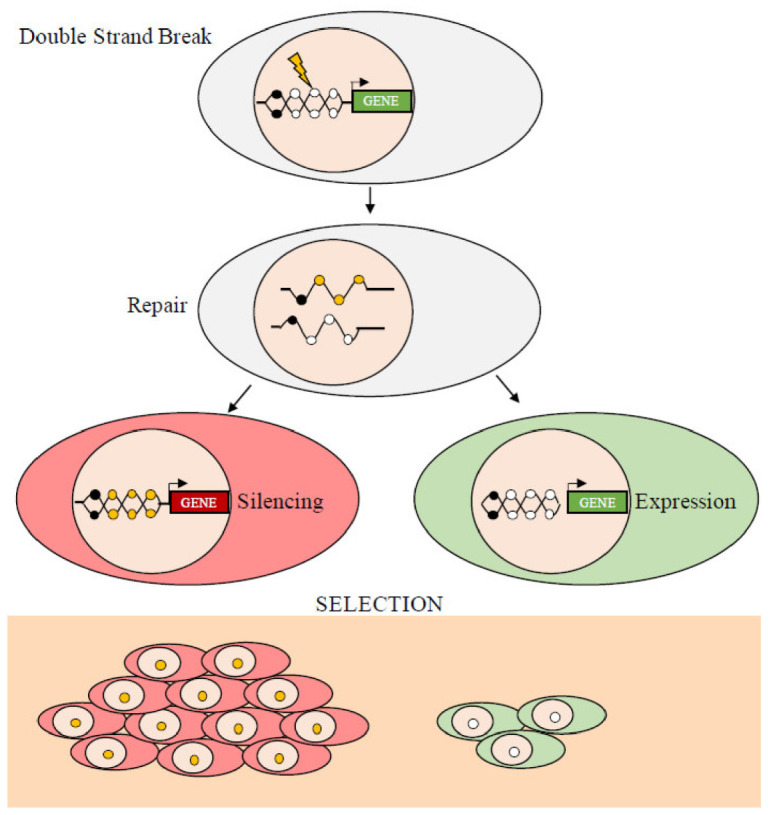
Double strand break (DSB) and homologous repair (HR) edit local methylation. Schematic representation of the events leading to the silencing or expression of DNA segments following DBS and HR. The yellow circles represent de novo methylated CpGs in HR clones. The black circles represent previously methylated CpGs. DNA damage and subsequent methylation are random, while gene silencing depends on the location of the methyl Cs. If the expression of the repaired gene is harmful, only cells that inherit the muted copy will survive. Conversely, if the function of the repaired gene is beneficial, the cells inheriting the non-methylated copy will have a growth advantage.

**Figure 4 cancers-13-01800-f004:**
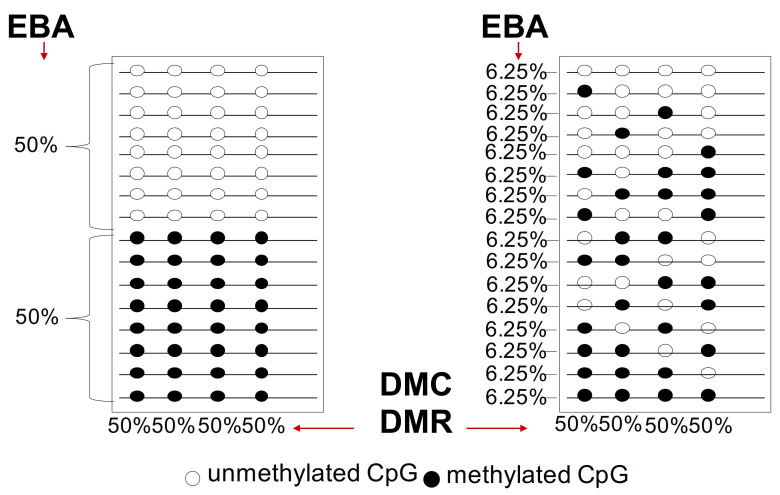
Differential methylated regions (DMC) and epiallele-based (EBA) analysis. Two populations of epialleles are reported on the right and left: the first population on the left is composed by two subpopulations equally represented (one fully methylated at the 4 CpGs and the other unmethylated). The second population on the right is composed by epialleles with differentially heterogeneous methylated CpGs with the same frequency 6.25%.

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
