# Peer review of "Epigenome Chaos: Stochastic and Deterministic DNA Methylation Events Drive Cancer Evolution"

_cancers, 2021, doi:10.3390/cancers13081800_

Round 1

Reviewer 1 Report

In this Review, Russo and colleagues discussed both genome and epigenome chaos of cancer, and focused on aberrant DNA methylation in cancer evolution. This review paper provides a novel concept of “epigenome chaos”, which summarized the epigenetic alterations observed in cancers. While this manuscript is well presented in general, the reviewer has following suggestions.

  1. In Figure 2, one light blue bar should be labeled as Intron;
  2. In Figure 4, it is confusing that “CpG” and “methylated CpG” are using the same symbol. Authors may remove “CpG”. In right panel of Figure 4B, sum of the values is 16% (1%*16). It could make more sense if the sum of the values is 100%, the same as left panel.
  3. Font size different than the rest of the manuscript in some lines, 85-101 and 221-226.
  4. In lines 91-101, the authors discussed clonal hematopoiesis of indeterminate potential, which is a very interesting point of view in terms of cancer evolution. However, the authors missed that this is also a great link for cancer evolution and epigenome chaos, since the most frequently mutated genes fall into the categories of epigenetic regulators (DNMT3a, TET2, and ASXL1) (e.g., PMID: 27215596). Please discuss it in the paper.

Author Response

Thank you for the careful handling of the manuscript submitted to cancers-1150921 " Epigenome Chaos: Stochastic and Deterministic DNA Methylation Events Drive Cancer Evolution."

We have corrected the errors expanded some sections.

In this Review, Russo and colleagues discussed both genome and epigenome chaos of cancer, and focused on aberrant DNA methylation in cancer evolution. This review paper provides a novel concept of “epigenome chaos”, which summarized the epigenetic alterations observed in cancers. While this manuscript is well presented in general, the reviewer has following suggestions.

1-In Figure 2, one light blue bar should be labeled as Intron;

Response 1: We have corrected the error.

2-In Figure 4, it is confusing that “CpG” and “methylated CpG” are using the same symbol. Authors may remove “CpG”. In right panel of Figure 4B, sum of the values is 16% (1%*16). It could make more sense if the sum of the values is 100%, the same as left panel.

Response 2: We have removed the reference to methylated and non-methylated CpG from panel A by moving it to panel B.

 3-Font size different than the rest of the manuscript in some lines, 85-101 and 221-226.

Response 3: We have corrected it.

4-In lines 91-101, the authors discussed clonal hematopoiesis of indeterminate potential, which is a very interesting point of view in terms of cancer evolution. However, the authors missed that this is also a great link for cancer evolution and epigenome chaos, since the most frequently mutated genes fall into the categories of epigenetic regulators (DNMT3a, TET2, and ASXL1) (e.g., PMID: 27215596). Please discuss it in the paper.

Response 4: We have added this link in the paragraph (line 113-126)

Reviewer 2 Report

Paper title: “Epigenome Chaos: Stochastic and Deterministic DNA Methylation Events Drive Cancer Evolution.” By Russo et al.

Comments:

-Figure 1 is too general. It just shows heterogeneity of cells but it's so generic that it doesn't add anything meaningful the way it stands. It's a good concept to get across, the authors should redo the figure showing how specific epigenetic changes leads to the selection of clones over time with cell division/tumor evolution. It will also be necessary to incorporate the role of genetic changes.

-The author talks about the Darwinian selection of genetic and epigenetic changes. This part needs more explanation of what they mean by that.

- In this context, the concept of the epigenetic driver of tumorigenesis and tumor growth is very relevant and important to discuss. I see the authors omitted this concept. They should include a section on the epigenetic driver of tumor and metastasis (This paper provided a framework for this idea: https://www.sciencedirect.com/science/article/pii/S1044579X17300536)

- They used the term Epigenetic Choas. I am not sure if this is useful at all. What does Chaos means? Cells are regulated tightly by the program, just because we don't understand it doesn't mean it's Chaos. I would like the authors to consider replacing this term and an alternative term could be Epigenetic stochasticity.

The section on Epialleles Based Analysis (EBA) is very superficial. The way it stands now is not useful. They should expand this section. Also, it's important to discuss in what situation of cancer epigenetics studies each of the tools should be used. Further, a reference or so that demonstrates the application of these tools should be mentioned and cited.

Author Response

Thank you for the careful handling of the manuscript submitted to cancers-1150921 " Epigenome Chaos: Stochastic and Deterministic DNA Methylation Events Drive Cancer Evolution."

We realize that the main concern of the referees was the concept of Epigenome Chaos and the link to genomic instability. There were also a number of errors in the figures and / or text.

We have corrected the errors and added a definition of chaos. Furthermore, following the advice of the referees, we have expanded some sections.

1 -Figure 1 is too general. It just shows heterogeneity of cells but it's so generic that it doesn't add anything meaningful the way it stands. It's a good concept to get across, the authors should redo the figure showing how specific epigenetic changes leads to the selection of clones over time with cell division/tumor evolution. It will also be necessary to incorporate the role of genetic changes.

Response 1: We have modified the figure including the effects of genetic and epigenetic variants on the cancer evolution.

2 -The author talks about the Darwinian selection of genetic and epigenetic changes. This part needs more explanation of what they mean by that.

Response 2: We have clearly stated that the both epigenetic and genetic changes can be randomly generated by they will be under deterministic selection based on fitness (line 68-72)

3 - In this context, the concept of the epigenetic driver of tumorigenesis and tumor growth is very relevant and important to discuss. I see the authors omitted this concept. They should include a section on the epigenetic driver of tumor and metastasis (This paper provided a framework for this idea: https://www.sciencedirect.com/science/article/pii/S1044579X17300536)

Response 3: The review focuses on epigenetic heterogeneity and its impact on the selection of clones in cancer. Following this advice, we added both the preneoplastic effect of mutations in genes associated with epigenetic control (line 113-120) and that the epigenetic alterations balance the activation of oncogenes favoring metastasis. (line 140-142).

4 - They used the term Epigenetic Chaos. I am not sure if this is useful at all. What does Chaos means? Cells are regulated tightly by the program, just because we don't understand it doesn't mean it's Chaos. I would like the authors to consider replacing this term and an alternative term could be Epigenetic stochasticity.

Response 4: We have revised this paragraph (line 46-53) specifying what we mean with epigenetic and genetic chaos:

“The evolution of cancer clones can be explained by the chaos theory, a concept borrowed from non-linear dynamic systems which evolve with apparent randomness but there are “underlying patterns, interconnectedness, constant feedback loops, repetition, self-similarity, fractals, and self-organization”.(Alcin, M. The Runge Kutta-4 based 4D Hy-perchaotic System Design for Secure Communication Applications. Chaos Theory and Applications, 2(1), 23-30, 202).

In fact methylation changes during tumor evolution are aperiodic, repetitive and very sensitive to the initial states of the system (ex. cell type or differentiation state). The genetic and epigenetic changes in evolving cancer cells are randomly generated but are selected by deterministic events (for example chemoresistance)”.

5 -The section on Epialleles Based Analysis (EBA) is very superficial. The way it stands now is not useful. They should expand this section. Also, it's important to discuss in what situation of cancer epigenetics studies each of the tools should be used. Further, a reference or so that demonstrates the application of these tools should be mentioned and cited.

Response 5: We have expanded the section. However, the choice of the tool to be used for the epigenetic analysis depends on the initial input (single locus or RRBS) and on the type of analysis to be applied (Entropy, Evolutionary Distance or identification of common signature) as described in the paragraph (line 225-257).

Round 2

Reviewer 2 Report

NA.

Check English and spelling for the final version.

Author Response

We would like to thank the reviewer for the pertinent comments which have helped us to improve the quality of our manuscript. Also, grammatical errors have been corrected.